# Development of Stilbenoid and Chalconoid Analogues as Potent Tyrosinase Modulators and Antioxidant Agents

**DOI:** 10.3390/antiox11081593

**Published:** 2022-08-17

**Authors:** Argyro Vontzalidou, Sapfo-Maria Dimitrakoudi, Konstantinos Tsoukalas, Grigoris Zoidis, Eliza Chaita, Evanthia Dina, Christina Cheimonidi, Ioannis P. Trougakos, George Lambrinidis, Alexios-Leandros Skaltsounis, Emmanuel Mikros, Nektarios Aligiannis

**Affiliations:** 1Division of Pharmacognosy and Natural Product Chemistry, Department of Pharmacy, National & Kapodistrian University of Athens, Panepistimiopolis Zografou, GR-15771 Athens, Greece; 2Division of Pharmaceutical Chemistry, Department of Pharmacy, National & Kapodistrian University of Athens, Panepistimiopolis Zografou, GR-15771 Athens, Greece; 3Department of Cell Biology and Biophysics, Faculty of Biology, National & Kapodistrian University of Athens, GR-15784 Athens, Greece

**Keywords:** tyrosinase inhibition, tyrosinase activation, free-radical-scavenging activity, diarylpropanes, diarylpropenoic acids, dihydrostilbenes, B16F1 and B16F10 cells, DPPH, ABTS

## Abstract

A number of stilbenoid and chalconoid derivatives were prepared by straightforward methods, and their ability to modulate tyrosinase activity and to scavenge free radicals were evaluated in vitro. The cell-free in vitro evaluation revealed two diarylpropanes, **24** and **25**, as potent tyrosinase inhibitors, whereas diarylpropenoic acids seemed to enhance the enzymatic activity. An in silico evaluation of the binding affinity of the selected compounds with the crystal structure of tyrosinase was also conducted in order to obtain better insight into the mechanism. Representative synthetic compounds with inhibitory and activating properties were further evaluated in melanoma cell lines B16F1 and B16F10 for their ability to moderate tyrosinase activity and affect melanin production. Dihydrostilbene analogues **I** and **II**, exhibited a stronger anti-melanogenic effect than kojic acid through the inhibition of cellular tyrosinase activity and melanin formation, while diarylpropanoic acid **44** proved to be a potent melanogenic factor, inducing cellular tyrosinase activity and melanin formation. Moreover, the antioxidant evaluation disclosed two analogues (**29** and **11**) with significant free-radical-scavenging activity (12.4 and 20.3 μM), which were 10- and 6-fold more potent than ascorbic acid (122.1 μΜ), respectively.

## 1. Introduction

Melanin is a dark pigment that defines various phenotypic features such as skin, hair and eye color [1], and is responsible for protecting the skin from UV radiation and oxidative stress. However, its atypical distribution in the skin may result in various esthetic problems and dermatological disorders, affecting a large part of the population. Hyperpigmentation disorders such as melanoma and pigmented patches can appear due to abnormal melanin accumulation, whereas vitiligo, a major depigmentation disorder, is observed as a result of its abnormal loss [2,3,4,5,6]. Melanin is produced in melanocytes [7], which derive from melanoblast precursor cells through a perplex process called melanogenesis. Melanoblasts are formed in the neural crest during embryogenesis and migrate to the skin, hair follicles, eyes, and ears during embryonic development [8,9]. Melanogenesis is regulated by several enzymatic and chemical reactions, but tyrosinase is regarded as the key enzyme, catalyzing the hydroxylation of tyrosine to 3,4-dihydroxyphenylalanine (L-DOPA) and finally to dopaquinone, by subsequent oxidation. Thus, modulating tyrosinase activity is considered as the major therapeutic target in treating skin pigmentation disorders. Moreover, tyrosinase catalyzes the formation of oxidation products (quinone derivatives), relating it to the demise of neurons in several neurodegenerative disorders such as Parkinson’s and Huntington’s disease [10,11,12]. However, it is also referred that tyrosinase contributes to the formation of neuromelanin (NM), which has been attributed to neuroprotective properties. Therefore, whether tyrosinase is beneficial or detrimental to neurons remains a bit controversial [13]. Furthermore, the increased production of free radicals and their harmful effect on vital biomolecules of the human body has been associated with the premature aging of the skin and the occurrence of age-related disorders such as Parkinson’s disease. Overall, the study of tyrosinase and free-radical-scavenging properties can be a useful and important target for the development of novel therapeutic agents or cosmeceuticals for several dermatological disorders and neurodegenerative diseases.

Following our continuing research on the discovery of novel tyrosinase inhibitors [14,15], we report the synthesis of a series of natural product analogues and their evaluation towards tyrosinase activity. Previous in silico studies combined with the in vitro evaluations of tyrosinase inhibition showed that dihydrostilbene analogues possess strong anti-tyrosinase activity. Dihydrostilbene analogues exhibited improved inhibitory activity compared to their deoxybenzoin precursors. More specifically, 2,4,4′-trihydroxy-dihydrostilbene I (Figure 1) was characterized as the most potent inhibitor (IC50 = 8.44 μΜ), more effective than the known inhibitor kojic acid (IC50 = 9.66 μΜ) [14]. These results encouraged us to proceed to the design and synthesis of new analogues bearing structural similarities to the dihydrostilbene scaffold, in order to explore the structural requirements for optimal activity. For this reason, in total 49 structural analogues were synthesized—12 diarylpropenoic acids, 5 diarylpropanoic acids, 2 3-arylcoumarins, 15 chalcones, 2 dihydrochalcones, 12 diarylpropanes and 1 diarylpropanol derivative—introducing a variety of substituents at the two aromatic rings and different functional groups in the middle chain of the scaffolds (Figure 1). All the obtained compounds were evaluated in vitro for their tyrosinase-inhibition properties, and in silico experiments were performed to investigate the binding affinity of the selected compounds with tyrosinase. In addition, representative compounds from the above-mentioned categories were evaluated for their ability to scavenge the free radicals DPPH and ABTS. Finally, the cytotoxicity of two tyrosinase inhibitors and two tyrosinase activators was assessed in melanoma cell lines (B16F1/B16F10), as well as their ability to moderate tyrosinase activity and affect melanin production (Table 1).

## 2. Materials and Methods

### 2.1. Experimental Procedures

#### 2.1.1. Chemistry

All chemicals and mushroom tyrosinase were purchased from Sigma-Aldrich Chemical Co. (Burlington, MA, United States). Nuclear magnetic resonance (NMR) spectra were obtained on a Bruker 600 MHz spectrometer using Me_2_CO-*d*_6_ and CDCl_3_ as solvents. The 2D-NMR experiments (COSY, HSQC and HMBC) were performed using standard Bruker microprograms. Absorption measurements were carried out on a Tecan (Infinite M200 PRO) plate reader. For the purification of the synthetic products, column chromatography was carried out using silica gel (Merck, Darmstadt, Germany) 0.04–0.06 mm (flash) with an applied pressure of 300 mbar. Precoated TLC silica 60 F254 plates (purchased from Merck, Darmstadt, Germany) were used for thin-layer chromatography (0.25). Spots were visualized using UV light and vanillin–sulfuric acid reagent. Elemental analyses (C, H, N) were performed by the Service Central de Microanalyse at CNRS (3 rue Michel-Ange, 75 016 Paris, FRANCE ) and were within ±0.4% of the theoretical values. The elemental analysis results for the tested compounds correspond to >95% purity.

#### 2.1.2. General Synthetic Procedures for Compounds

##### General Synthetic Procedure for Chalcones

A 3.39 mL aqueous solution of potassium hydroxide (50%) was added dropwise at room temperature to a solution of the appropriate substituted acetophenone (4.76 mmol) and the appropriate benzaldehyde (5.48 mmol) in ethanol (40 mL), and the mixture was stirred at 90 °C for 4–6 h (TLC monitoring). The solution was then neutralized with an aqueous solution of HCl (10%). After the addition of water (80 mL) the reaction mixture was sequentially extracted with EtOAc (3 × 100 mL). The combined organic layers were washed with H_2_O (3 × 70 mL) and brine (2 × 70 mL), dried over anh. Na_2_SO_4_ and evaporated in vacuo. The crude residue was purified by flash column chromatography on silica gel using a mixture of c-Hexane/AcOEt with increasing polarity to afford the respective chalcones 1–15 (yield 60–92%) as solids.

##### (*E*)-1-(4-(Benzyloxy)phenyl)-3-(4-methoxyphenyl)prop-2-en-1-one (**1**)

White solid, 86% yield. ^1^H-NMR: (600 MHz, CDCl_3_) δ: 8.03 (2H, d, *J* = 8.9 Hz, H-2′/H-6′), 7.78 (1H, d, *J* = 15.6 Hz, H-β), 7.60 (2H, d, *J* = 8.7 Hz, H-2/H-6), 7.44 (4H, m, H-2″/H-6″ και H-3″/H-5″), 7.43 (1H, d, *J* = 15.6 Hz, H-α), 7.38 (1H, d, *J* = 8.2 Hz, H-4″), 7.05 (2H, d, *J* = 8.9 Hz, H-3′/H-5′), 6.94 (2H, d, *J* = 8.8 Hz, H-3/H-5), 5.19 (2H, s, -CH_2_), 3.84 (3H, s, -OCH_3_); ^13^C-NMR: (150 MHz, CDCl_3_) δ: 188.8 (C=O), 162.5 (C-4′), 161.1 (C-4), 144.0 (C-β), 138.6 (C-1″), 131.4 (C-2″/C-6″), 130.8 (C-2′/C-6′), 130.4 (C-1), 130.0 (C-2/C-6), 128.6 (C-4″), 127.6 (C-3″/5″), 119.6 (C-α), 114.7 (C-3′/C-5′), 114.3 (C-3/C-5), 70.1 (-CH_2_), 53.1 (-OCH_3_). 

##### (*E*)-1,3-Bis(4-methoxyphenyl)prop-2-en-1-one (**2**)

White solid, 76% yield. ^1^H-NMR: (600 MHz, CDCl_3_) δ: 8.03 (2H, d, *J* = 8.8 Hz, H-2′/H-6′), 7.78 (1H, d, *J* = 15.6 Hz, H-β), 7.60 (2H, d, *J* = 8.7 Hz, H-2/H-6), 7.43 (1H, d, *J* = 15.6 Hz, H-α), 6.98 (2H, d, *J* = 8.8 Hz, H-3′/H-5′), 6.93 (2H, d, *J* = 8.7 Hz, H-3/H-5), 3.88 (3H, s, 4′-OCH_3_), 3.85 (3H, s, 4-OCH_3_); ^13^C-NMR: (150 MHz, CDCl_3_) δ: 189.0 (C=O), 163.2 (C-4′), 161.3 (C-4), 143.9 (C-β), 136.0 (C-1), 131.1 (C-2′/C-6′), 130.4 (C-1′), 130.0 (C-2/C-6), 119.5 (C-α), 114.5 (C-3/C-5), 114.0 (C-3′/C-5′), 55.4 (2x-OCH_3_).

##### (*E*)-3-(4-(Benzyloxy)phenyl)-1-(4-methoxyphenyl)prop-2-en-1-one (**3**)

White solid, 93% yield. ^1^H-NMR: (600 MHz, CDCl_3_) δ: 8.04 (2H, d, *J* = 8.8 Hz, H-2′/H-6′), 7.78 (1H, d, *J* =15.6 Hz, H-β), 7.60 (2H, d, *J* =8.7 Hz, H-2/H-6), 7.44 (2H, d, *J* =7.4 Hz, H-2″/H-6″), 7.43 (1H, d, *J* =15.6 Hz, H-α), 7.41 (2H, t, *J* = 7.4 Hz, H-3″/H-5″), 7.34 (1H, m, H-4″), 7.01 (2H, d, *J* = 8.7 Hz, H-3/H-5), 6.98 (2H, d, *J* = 8.8 Hz, H-3′/H-5′), 5.12 (2H, s, -CH_2_), 3.89 (3H, s, -OCH_3_); ^13^C-NMR: (150 MHz, CDCl_3_) δ: 188.2 (C=O), 163.1 (C-4′), 160.3 (C-4), 143.6 (C-β), 136.2 (C-1″), 131.3 (C-1′), 130.7 (C-2′/C-6′), 130.2 (C-2/C-6), 128.7 (C-3″/C-5″), 128.4 (C-4″), 127.8 (C-1), 127.4 (C-2″/C-6″), 119.7 (C-α), 115.3 (C-3/C-5), 113.8 (C-3′/C-5′), 70.1 (-CH_2_), 55.5 (-OCH_3_).

##### (*E*)-1-(4-(Benzyloxy)phenyl)-3-(2-chlorophenyl)prop-2-en-1-one (**4**)

White solid, 87% yield. ^1^H-NMR: (600 MHz, CDCl_3_) δ: 8.17 (1H, d, *J* = 15.7 Hz, H-β), 8.04 (2H, d, *J* = 8.7 Hz, H2′/6′), 7.74 (1H, dd, *J* = 6.8/2.0 Hz, H-6), 7.50 (1H, d, *J* = 15.7 Hz, H-α), 7.45 (2H, d, J = 7.1 Hz, H-2″/H-6″), 7.43 (1H, d, *J* = 7.2 Hz, H-3), 7.41 (2H, t, *J* = 7.1 Hz, H-3″/H-5″), 7.35 (1H, d, *J* = 7.1 Hz, H-4″), 7.32 (2H, m, H-4/H-5), 5.16 (2H, s, CH_2_); ^13^C-NMR: (150 MHz, CDCl_3_) δ: 188.5 (C=O), 162.5 (C-4′), 139.7 (C-β), 139.7 (C-2), 136.2 (C-1″), 133.4 (C-1), 130.7 (C-2′/C-6′), 130.7 (C-4), 130.1 (C-3), 128.3 (C-3″/C-5″), 128.0 (C-4″), 127.5 (C-6), 127.2 (C-2″/6″), 126.7 (C-5), 124.7 (C-α), 114.6 (C-3′/C-5′), 69.8 (CH_2_). 

##### (*E*)-1-(4-(Benzyloxy)phenyl)-3-(2,4-dichlorophenyl)prop-2-en-1-one (**5**)

White solid, 87% yield. ^1^H-NMR: (600 MHz, CDCl_3_) δ: 8.08 (1H, d, *J* = 15.7 Hz, H-β) 8.03 (2H, d, *J* = 8.8 Hz, H-2′/H-6′), 7.67 (1H, d, *J* = 8.5 Hz, H-6), 7.48 (1H, d, *J* = 15.7 Hz, H-α), 7.44 (2H, d, H-2″/H-6″), 7.43 (1H, s, H-3), 7.41 (1H, d, *J* = 7.4 Hz, H3″/5″), 7.35 (1H, m, H-4″), 7.30 (2H, d, *J* = 8.5/2 Hz, H-5), 7.06 (2H, d, *J* = 8.8 Hz, H3′/5′), 5.16 (2H, s, -CH_2_); ^13^C-NMR: (150 MHz, CDCl_3_) δ: 188.1 (C=O), 162.5 (C-4′), 138.9 (C-β), 136.2 (C-1″), 136.0 (C-2 /C-4), 133.4 (C-1), 131.1 (C-2′/C-6′), 130.7 (C-1′), 130.1 (C-2″/6″), 130.0 (C-3), 128.7 (C-6), 128.4 (C-3″/C-5″), 128.2 (C-4″), 127.7 (C-5), 124.8 (C-α), 114.8 (C-3′/C-5′), 70.2 (CH_2_).

##### (*E*)-3-(2-Chlorophenyl)-1-(4-methoxyphenyl)prop-2-en-1-one (**6**)

White solid, 88% yield. ^1^H-NMR: (600 MHz, CDCl_3_) δ: 8.16 (1H, d, *J* = 15.7 Hz, H-β), 8.03 (2H, d, *J* = 8.7 Hz, H-2′/H-6′), 7.74 (1H, dd, *J* = 6.8/2.2 Hz, H-6), 7.49 (1H, d, *J* = 15.7 Hz, H-α), 7.43 (1H, d, *J* = 6.8/2.2 Hz, H-3), 7.31 (2H, m, H-4/H-5), 6.98 (2H, d, *J* = 8.7 Hz, H-3′/H-5′), 3.89 (3H, s, -OCH_3_); ^13^C-NMR: (150 MHz, CDCl_3_) δ: 188.5 (C=O), 163.7 (C-4′), 139.8 (C-2), 139.7 (C-β), 135.3 (C-1), 130.8 (C-2′/C-6′), 130.8 (C-5), 130.5 (C-1′), 130.2 (C-3), 127.6 (C-6), 126.8 (C-4), 124.5 (C-α), 113.6 (C-3′/C-5′), 55.4 (-OCH_3_).

##### (*E*)-3-(2,4-Dichlorophenyl)-1-(4-methoxyphenyl)prop-2-en-1-one (**7**)

White solid, 90% yield. ^1^H-NMR: (600 MHz, CDCl_3_) δ: 8.07 (1H, d, *J* = 15.7 Hz, H-β), 8.02 (2H, d, *J* = 8.7 Hz, H-2′/H-6′), 7.67 (1H, d, *J* = 8.4 Hz, H-6), 7.47 (1H, d, *J* = 15.7 Hz, H-α), 7.45 (1H, d, *J* = 1.9 Hz, H-3), 7.29 (1H, dd, *J* = 8.4/1.9 Hz, H-5), 6.98 (2H, d, *J* = 8.7 Hz, H-3′/H-5′), 3.89 (3H, s, OCH_3_); ^13^C-NMR: (150 MHz, CDCl_3_) δ: 188.1 (C=O), 163.2 (C-4′), 138.7 (C-β), 138.6 (C-4), 136.0 (C-1), 132.0 (C-2), 131.1 (C-2′/C-6′), 130.4 (C-1′), 130.1 (C-3), 128.7 (C-6), 127.6 (C-5), 125.0 (C-α), 114.0 (C-3′/C-5′), 55.3 (-OCH_3_). 

##### (*E*)-1,3-Bis(4-(benzyloxy)phenyl)prop-2-en-1-one (**8**)

White solid, 90% yield. ^1^H-NMR: (600 MHz, CDCl_3_) δ: 8.04 (2H, d, *J* = 8.7 Hz, H2′/6′), 7.79 (1H, d, *J* = 15.8 Hz, H-β), 7.61 (2H, d, *J* = 8.7 Hz, H-2/H-6), 7.44 (8H, m, H-2″/H-6″, H-3″/H-5″, H-2″′/H-6″′ και H-3″′/H-5″′), 7.43 (1H, d, *J* = 15.8 Hz, H-α), 7.37 (2H, dd, *J* = 8.7 Hz, H-4″ και H-4″′), 7.07 (2H, d, *J* = 8.7 Hz, H-3′/H-5′), 7.02 (2H, d, *J* = 8.7 Hz, H-3/H-5), 5.16 (2H, s, -CH_2_), 5.13 (2H, s, -CH_2_).

##### (*E*)-1-(4-(Benzyloxy)phenyl)-3-(3,4,5-trimethoxyphenyl)prop-2-en-1-one (**9**)

White solid, 15% yield. ^1^H-NMR: (600 MHz, CDCl_3_) δ: 8.03 (2H, d, *J =* 8.8 Hz, H-2′/H-6′), 7.71 (1H, d, *J =* 15.5 Hz, H-β), 7.44 (2H, d, *J =* 7.4 Hz, H-2″/H-6″), 7.42 (1H, d, *J =* 15.5 Hz, H-α), 7.40 (2H, d, *J =* 7.4 Hz, H-3″/H-5″), 7.36 (1H, d, *J =* 7.4 Hz, H-4″), 6.86 (2H, s, H-2/H-6), 5.16 (2H, s, -CH_2_), 3.92 (6H, s, 3,5-OCH_3_), 3.90 (3H, s, 4-OCH_3_); ^13^C-NMR: (150 MHz, CDCl_3_) δ: 188.3 (C=O), 162.5 (C-4′), 153.1 (C-3/C-5), 143.9 (C-β), 140.0 (C-4), 136.3 (C-1″), 131.4 (C-1′), 131.0 (C-1), 130.5 (C-2′/C-6′), 128.4 (C-3″/C-5″), 128.1 (C-4″), 127.3 (C-2″/C-6″), 121.0 (C-α), 114.5 (C-3′/C-5′), 105.3 (C-2/6), 69.9 (-CH_2_), 60.7 (4- OCH_3_), 55.7 (3,5-OCH_3_).

##### (*E*)-1-(2,4-Dihydroxyphenyl)-3-(4-hydroxyphenyl)prop-2-en-1-one (**10**)

White solid, 60% yield. ^1^H-NMR: (600 MHz, Me_2_CO-*d*_6_) δ: 13.63 (1H, s, 2′-OH), 9.67 (1H, brs, 4-OH), 9.16 (1H, brs, 4′-OH), 8.11 (1H, d, *J* = 8.5 Hz, H-6′), 7.83 (1H, d, *J* = 15.3 Hz, H-β), 7.76 (1H, d, *J* = 15.3Hz, H-α), 7.74 (2H, d, *J* = 8.4 Hz, H-2/H-6), 6.93 (2H, d, *J* = 8.4 Hz, H-3/H-5), 6.64 (1H, d, *J* = 8.5 Hz, H-5′), 6.36 (1H, s, H-3′); ^13^C-NMR: (150 MHz, Me_2_CO-*d*_6_) δ: 193.8 (C=O), 167.6 (C-2′), 166.7 (C-4′), 161.8 (C-4), 146.1 (C-β), 134.1 (C-6′), 132.6 (C-2/C-6), 128.3 (C-1), 119.1 (C-α), 117.6 (C-3/C-5), 115.6 (C-1′), 109.6 (C-5′), 104.6 (C-3′).

##### (*E*)-1-(2,4-Dihydroxyphenyl)-3-(3,4,5-trimethoxyphenyl)prop-2-en-1-one (**11**)

White solid, 60% yield. ^1^H-NMR: (600 MHz, CDCl_3_) δ: 13.36 (1H, s, 2′-OH), 7.84 (1H, d, *J =* 8.8 Hz, H-6′), 7.81 (1H, d, *J =* 15.0 Hz, H-β), 6.87 (2H, s, H-2″/H-6″), 6.45 (1H, d, *J* = 3.0 Hz, H-3′), 6.34 (1H, dd, *J* = 8.4, 2.0 Hz, H-5′), 3.94 (6H, s, 3″,5″-OCH_3_), 3.91 (3H, s, 4″-OCH_3_); ^13^C-NMR: (150 MHz, CDCl_3_) δ: 191.6 (C=O), 165.9 (C-2′), 164.9 (C-4′), 153.4 (C-3″/C-5″), 144.2 (C-*β*), 140.3 (C-4″), 131.9 (C-6′), 130.4 (C-1″), 119.7 (C-α), 113.5 (C-1′), 108.4 (C-2″/6″), 105.8 (C-5′), 103.1 (C-3′), 60.9 (4″- OCH_3_), 56.2 (3″,5″-OCH_3_).

##### (*E*)-1-(4-(Benzyloxy)-2-hydroxyphenyl)-3-(3,4,5-trimethoxyphenyl)prop-2-en-1-one (**12**)

White solid, 25% yield. ^1^H-NMR: (600 MHz, CDCl_3_) δ: 13.44 (1H, s, 2′-OH), 7.84 (1H, d, *J =* 9.6 Hz, H-6′), 7.81 (1H, d, *J =* 15.6 Hz, H-β), 7.42 (5H, m, Bn-H), 6.87 (2H, s, H-2″/H-6″), 6.58 (2H, m, H-3′,5′), 5.13 (2H, s, CH_2_), 3.94 (6H, s, 3″,5″-OCH_3_), 3.91 (3H, s, 4″-OCH_3_); ^13^C-NMR: (150 MHz, CDCl_3_) δ: 193.1 (C=O), 165.5 (C-2′), 162.9 (C-4′), 153.3 (C-3″/C-5″), 143.3 (C-*β*), 140.7 (C-4″), 131.3 (C-6′), 130.7 (C-1″), 120.5 (C-α), 114.6 (C-1′), 108.2 (C-5′), 106.3 (C-2″/6″), 101.4 (C-3′), 70.1 (-CH_2_), 60.7 (4″- OCH_3_), 56.2 (3″,5″-OCH_3_).

##### (*E*)-3-(3-Hydroxyphenyl)-1-(3,4,5-trimethoxyphenyl)prop-2-en-1-one (**13**)

White solid, 69% yield. ^1^H-NMR: (600 MHz, CDCl_3_) δ: 7.74 (1H, d, *J =* 15.6 Hz, H-β), 7.67 (1H, vbrs, OH), 7.43 (1H, d, *J =* 15.6 Hz, H-α), 7.25–7.21 (3H, m, H-4,5,6), 7.16 (2H, s, H-2′,6′), 6.95 (1H, m, H-4), 3.94 (6H, s, 4-OCH_3_), 3.91 (6H, s, 3,5-OCH_3_); ^13^C-NMR: (150 MHz, CDCl_3_) δ: 190.5 (C=O), 159.1 (C-3), 153.1 (C-3′,5′), 144.4 (C-β), 142.6 (C-4), 136.5 (C-1), 134.6 (C-5), 130.2 (C-1), 121.6 (C-6), 121.6 (C-α), 117.3 (C-5), 115.4 (C-2), 105.9 (C-2′,6), 60.6 (4- OCH_3_), 56.4 (3,5-OCH_3_).

##### (*E*)-1-(2,4,6-Trimethoxyphenyl)-3-(3,4,5-trimethoxyphenyl)prop-2-en-1-one (**14**)

White solid, 93% yield. ^1^H-NMR: (600 MHz, CDCl_3_) δ: 7.27 (1H, d, *J =* 15.6 Hz, H-β), 6.89 (1H, d, *J =* 15.6 Hz, H-α), 6.79 (2H, s, H-2/H-6), 6.20 (2H, s, H-3′/H-5′), 3.90 (12H, s, 3,4,5,4′-OCH_3_), 3.80 (6H, s, 2′,6′-OCH_3_); ^13^C-NMR: (150 MHz, CDCl_3_) δ: 188.3 (C=O), 159.7 (C-2′,4′,6′), 153.5 (C-3,5), 145.0 (C-β), 138.5 (C-1), 136.7 (C-4), 128.5 (C-α), 111.8 (C-1′), 105.9 (C-2,6), 91.0 (C-3′/C-5′), 60.0 (4- OCH_3_), 56.0 (2′,6′-OCH_3_), 55.0 (3,5,4′-OCH_3_).

##### *(E*)-1-(4-Methoxyphenyl)-3-(3,4,5-trimethoxyphenyl)prop-2-en-1-one (**16**)

White solid, 79% yield. ^1^H-NMR: (600 MHz, CDCl_3_) δ: 7.92 (2H, d, *J =* 8.6 Hz, H-2′/H-6′), 7.71 (1H, d, *J =* 16.3 Hz, H-β), 7.41 (1H, d, *J =* 16.3 Hz, H-α), 6.87 (2H, s, H-2/H-6), 6.59 (2H, d, *J =* 8.6 Hz, H-3′/H-5′), 3.92 (6H, s, 3,5-OCH_3_), 3.90 (6H, s, 4,4′-OCH_3_); ^13^C-NMR: (150 MHz, CDCl_3_) δ: 189.6 (C=O), 163.8 (C-4′), 153.5 (C-3,5), 144.6 (C-β), 140.1 (C-4), 131.0 (C-2′,6′), 130.8 (C-1′), 130.2 (C-1), 121.6 (C-α), 115.0 (C-3′/C-5′), 105.9 (C-2/C-6), 61.0 (4- OCH_3_), 56.4 (3,5-OCH_3_), 55.6 (4′-OCH_3_).

General Synthetic Procedure for Dihydrochalcones

BF_3_-Et_2_O (12.5 mL) was added under nitrogen atmosphere to a mixture of the appropriate alcohol (5 mmol) and the appropriate phenylpropanoic acid (5 mmol). The solution was heated for 3–5 h (TLC monitoring) in an oil bath at 95 °C under argon and then poured into an ice–water mixture and extracted with ethyl acetate (3 × 100 mL). The combined organic layers were washed with H_2_O (3 × 70 mL) and brine (2 × 70 mL), dried over anh. Na_2_SO_4_ and evaporated in vacuo. The residue was purified by flash column chromatography on silica gel using c-Hexane/EtOAc as the eluents to afford the respective dihydrochalcones **16**–**17** in good yields (yield 60–70%) as crystalline solids.

##### 4,2′,4′-Trihydroxydihydrochalcone (**16**)

^1^H-NMR: (600 MHz, Me_2_CO-*d*_6_) δ: 12.83 (1H, s, 2′-OH), 9.56 (1H, s, 4-OH), 8.21 (1H, s, 4′-OH), 7.83 (1H, d, *J =* 8.8 Hz, H-6′), 7.13 (2H, d, *J =* 8.4 Hz, H-2/H-6), 6.77 (2H, d, *J =* 8.4 Hz, H-3/H-5), 6.35 (1H, d, *J =* 2.3 Hz, H-3′), 6.44 (1H, dd, *J =* 8.8/2.3 Hz, H-5′), 3.26 (2H, t, *J =* 7.7 Hz, H-α), 2.94 (2H, t, *J =* 7.7 Hz, H-β); ^13^C-NMR: (150 MHz, Me_2_CO-*d_6_*) δ: 205.8 (C=O), 166.1 (C-2′), 165.7 (C-4′), 156.9 (C-4), 134.4 (C-6′), 133.2 (C-1), 131.1 (C-2/C-6), 116.7 (C-3/C-5), 114.7 (C-1′), 109.4 (C-5′), 104.2 (C-3′), 40.9 (C-α), 30.8 (C-β).

##### 3-(4-Hydroxyphenyl)-1-(2,4,6-trimethoxyphenyl)propan-1-one (**17**)

White solid, yield: 64%, ^1^H-NMR: (600 MHz, CDCl_3_) δ: 7.07 (2H, d, *J =* 8.2 Hz, H-2″/H-6″), 6.73 (2H, d, *J =* 8.2 Hz, H-3″/H-5″), 6.09 (2H, s, H-3′, H-5′), 3.82 (3H, s, 4′-OCH_3_), 3.75 (6H, s, 2′/6′-OCH_3_), 3.02 (2H, t, *J =* 8.2 Hz, H-3), 2.92 (2H, t, *J =* 8.2 Hz, H-2); ^13^C-NMR: (150 MHz, CDCl_3_) δ: 203.4 (C=O), 162.1 (C-4′), 158.0 (C-2′/C-6′), 153.4 (C-4″), 133.7 (C-1″), 129.9 (C-2″/C-6″), 115.4 (C-3″/C-5″), 113.4 (C-1′), 90.9 (C-3′/C-5′), 56.0 (2′/6′-OCH_3_), 55.6 (4′-OCH_3_), 46.7 (C-2), 29.2 (C-3).

##### General Synthetic Procedure for Diarylpropanes (**18**–**24**)

A 10 wt % catalyst of 10% Pd/C was added to a solution of the appropriate chalcone (0.43 mmol) in EtOAc (30 mL). After 5 h of shaking under an atmosphere of 50–55 psi hydrogen at room temperature, the catalyst was removed by filtration on celite and washed with EtOAc. The combined filtrates were concentrated to dryness under reduced pressure to afford the corresponding diarylpropanes in almost quantitative yield.

##### 4-(3-(4-Methoxyphenyl)propyl)phenol (**18**)

^1^H-NMR: (600 MHz, CDCl_3_) δ: 7.11 (2H, d, *J* = 8.4 Hz, H-2″/H-6″), 7.05 (2H, d, *J* = 8.3 Hz, H-2′/H-6′), 6.84 (2H, d, *J* = 8.4 Hz, H-3″/H-5″), 6.76 (2H, d, *J* = 8.3 Hz, H-3′/H-5′), 4.61 (1H, s, -OH), 3.80 (3H, s, -OCH_3_), 2.58 (4H, m, H-1/H-3), 1.89 (2H, quin, *J* = 7.6 Hz, H-2); ^13^C-NMR: (150 MHz, CDCl_3_) δ: 157.5 (C-4″), 153.4 (C-4′), 134.5 (C-1″), 134.1 (C-1′), 129.1 (C-2″/6″), 129.1 (C-2′/6′), 114.4 (C-3′/5′), 114.1 (C-3″/5″), 54.8 (-OCH_3_), 34.1 (C-1/C-3), 33.1 (C-2).

##### 1,3-Bis(4-methoxyphenyl)propane (**19**)

^1^H-NMR: (600 MHz, CDCl_3_) δ: 7.11 (4H, d, *J* = 8.4 Hz, H-2′/H-6′ και H-2″/H-6″), 6.84 (4H, d, *J* = 8.4 Hz, H-3′/H-5′ και H-3″/H-5″), 3.80 (6H, s, 4′-OCH_3_/4″-OCH_3_), 2.59 (4H, t, *J* = 7.7 Hz, H-1/H-3), 1.90 (2H, quin, *J* = 7.7 Hz, H-2); ^13^C-NMR: (150 MHz, CDCl_3_) δ:157.4 (C-4′ και C-4″), 134.5 (C-1′ και C-1″), 129.3 (C-2′/C-6′ και C-2″/C-6″), 114.1 (C-3′/5′ και C-3″/5″), 53.3 (4′-OCH_3_/4″-OCH_3_), 34.4 (C-1/C-3), 33.5 (C-2).

##### 4-(3-(2-Chlorophenyl)propyl)phenol (**20**)

^1^H-NMR: (600 MHz, CDCl_3_) δ: 7.36 (1H, br d, *J* = 8.2 Hz, H-3″), 7.25–7.21 (3H, m, H-4″/H-5″/H-6″), 7.09 (2H, d, *J* = 8.4 Hz, H-2′/H-6′), 6.80 (2H, d, *J* = 8.4 Hz, H-3′/H-5′), 2.79 (2H, t, *J* = 7.7 Hz, H-3), 2.62 (2H, t, *J* = 7.7 Hz, H-1), 1.96 (2H, quin, *J* = 7.7 Hz, H-2); ^13^C-NMR: (150 MHz, CDCl_3_) δ:153.8 (C-4′), 139.8 (C-2″), 134.1 (C-1′), 133.8 (C-1″), 130.8 (C-5″), 129.5 (C-3″), 129.4 (C-2′/C-6′), 127.6 (C-6″), 126.8 (C-4″), 115.2 (C-3′/5′), 34.5 (C-1), 33.2 (C-3), 29.7 (C-2).

##### 4-(3-(2,4-Dichlorophenyl)propyl)phenol (**21**)

^1^H-NMR: (600 MHz, CDCl_3_) δ: 7.35 (1H, d, *J* = 2.0 Hz, H-3″), 7.16 (1H, dd, *J* = 8.2/2.0 Hz, H-5″), 7.12 (1H, d, *J* = 8.2 Hz, H-6″), 7.06 (2H, d, *J* = 8.4 Hz, H-2′/H-6′), 6.77 (2H, d, *J* = 8.4 Hz, H-3′/H-5′), 4.99 (1H, -OH), 2.71 (2H, t, *J* = 7.7 Hz, H-3), 2.61 (2H, t, *J* = 7.7 Hz, H-1), 1.89 (2H, t, *J* = 7.7 Hz, H-2); ^13^C-NMR: (150 MHz, CDCl_3_) δ: 153.7 (C-4′), 138.2 (C-1″), 134.5 (C-2″), 133.9 (C-1′), 132.0 (C-4″), 131.3 (C-6″), 129.8 (C-2′/C-6′), 129.5 (C-3″), 127.1 (C-5″), 115.5 (C-3′/C-5′), 34.8 (C-1), 32.8 (C-3), 31.5 (C-2).

##### 1-Chloro-2-(3-(4-methoxyphenyl)propyl)benzene (**21**)

^1^H-NMR: (600 MHz, CDCl_3_) δ: 7.37 (1H, br d, *J* = 8.2 Hz, H-3″), 7.25–7.21 (3H, m, H-4″/H-5″/H-6″), 7.16 (2H, d, *J* = 8.4 Hz, H-2′/H-6′), 6.87 (2H, d, *J* = 8.4 Hz, H-3′/H-5′), 3.84 (CH_3_O-4′), 2.80 (2H, t, *J* = 7.7 Hz, H-3), 2.66 (2H, t, *J* = 7.7 Hz, H-1), 1.97 (2H, quin, *J* = 7.7 Hz, H-2); ^13^C-NMR: (150 MHz, CDCl_3_) δ: 157.7 (C-4′), 139.8 (C-2″), 134.1 (C-1′), 133.8 (C-1″), 130.5 (C-5″), 129.5 (C-3″), 129.3 (C-2′/C-6′), 127.1 (C-6″), 126.8 (C-4″), 113.7 (C-3′/5′), 55.1 (CH_3_O-4′), 34.7 (C-1), 33.2 (C-3), 31.5 (C-2).

##### 2,4-Dichloro-1-(3-(4-methoxyphenyl)propyl)benzene (**23**)

^1^H-NMR: (600 MHz, CDCl_3_) δ: 7.35 (1H, d, *J* = 1.9 Hz, H-3″), 7.16 (1H, dd, *J* = 8.0/1.9 Hz, H-5″), 7.12 (1H, d, *J* = 8.0 Hz, H-6″), 7.11 (2H, d, *J* = 8.5 Hz, H2′/6′), 6.84 (2H, d, *J* = 8.5 Hz, H3′/5′), 3.80 (3H, s, -OCH_3_), 2.71 (2H, t, *J* = 7.8 Hz, H-3), 2.63 (2H, t, *J* = 7.8 Hz, H-1), 1.90 (2H, t, *J* = 7.8 Hz, H-2); ^13^C-NMR: (150 MHz, CDCl_3_) δ:157.3 (C-4′), 138.2 (C-1″), 134.1 (C-2″), 133.9 (C-1′), 132.0 (C-4″), 131.1 (C-6″), 129.5 (C-2′/6′), 129.3 (C-3″), 127.0 (C-5″), 113.9 (C-3′/5′), 55.3 (-OCH_3_), 34.7 (C-1), 32.7 (C-3), 31.5 (C-2).

##### 4-(3-(4-Hydroxyphenyl)propyl)benzene-1,3-diol (**24**)

^1^H-NMR: (600 MHz, Me_2_CO-*d*_6_) δ: 8.07 (1H, s, 4″-OH), 8.04 (1H, s, 2′-OH), 7.95 (1H, s, 4′-OH), 7.02 (2H, d, *J* = 8.3 Hz, H-2″/6″), 6.85 (1H, d, *J* = 8.1 Hz, H-6′), 6.73 (2H, d, *J* = 8.3 Hz, H3″/5″), 6.30 (1H, d, *J* = 2.2 Hz, H-3′), 6.26 (1H, dd, *J* = 8.1/2.2 Hz, H-5′), 2.54 (4H, t, *J* = 7.7 Hz, H-1 και H-3), 1.80 (2H, quin, *J* = 7.7 Hz, H-2); ^13^C-NMR: (150 MHz, Me_2_CO-*d*_6_) δ: 157.9 (C-4′), 157.0 (C-2′), 156.6 (C-4″), 134.7 (C-1″), 131.9 (C-6′), 130.8 (C-2″/6″), 116.7 (C-3″/5″), 108.1 (C-5′), 104.1 (C-3′), 36.1 (C-3), 33.9 (C-2), 30.7 (C-1).

##### 4-(3-(3,4,5-Trimethoxyphenyl)propyl)benzene-1,3-diol (**25**)

^1^H-NMR: (600 MHz, CDCl_3_) δ: 6.94 (1H, d, *J* = 8.2 Hz, H-6′), 6.41 (2H, s, H-2″/6″), 6.35 (1H, dd, *J* = 8.2, 2.4 Hz, 5′), 6.31 (1H, d, *J* = 2.4 Hz, H-3′), 3.84 (6H, s, 3″-OCH_3_/5″-OCH_3_), 3.83 (3H, s, 4″-OCH_3_), 2.61 (2H, t, *J* = 7.9 Hz, H-3), 2.56 (2H, t, *J* = 7.5 Hz, H-1), 1.91 (2H, quin, *J* = 7.6 Hz, H-2); ^13^C-NMR: (150 MHz, CDCl_3_) δ: 154.6 (C-4′/2′), 153.0 (C-3‘/5‘), 138.2 (C-1″), 135.8 (C-4″), 131.0 (C-6′), 107.7 (C-5′), 105.6 (C-2″/6″), 103.1 (C-3′), 61.0 (4″-OCH_3_), 56.2 (3″-OCH_3_/5″-OCH_3_), 35.9 (C-3), 31.4 (C-2), 28.9 (C-1).

##### 3-(3-(3,4,5-Trimethoxyphenyl)propyl)phenol (**26**)

^1^H-NMR: (600 MHz, CDCl_3_) δ: 7.15 (1H, t, *J* = 7.7 Hz, H-5″), 6.76 (1H, d, *J* =7.7 Hz, H-4″), 6.75 (1H, d, *J* = 7.6 Hz, H-6″), 6.68 (1H, brs, H-2″), 6.39 (2H, s, H-2′/6′), 3.84 (6H, s, 3′-OCH_3_/5′-OCH_3_), 3.83 (3H, s, 4′-OCH_3_), 2.60 (2H, t, *J* = 7.6 Hz, H-3), 2.60 (2H, t, *J* = 7.6 Hz, H-1), 1.93 (2H, quin, *J* = 7.6 Hz, H-2); ^13^C-NMR: (150 MHz, CDCl_3_) δ: 155.8 (C-3‘’), 153.3 (C-3‘/5‘), 143.6 (C-1″), 140.3 (C-1′), 137.3 (C-4‘), 129.8 (C-5‘’), 120.9 (C-6‘’), 113.0 (C-4″), 103.1 (C-2‘/6′), 61.0 (4″-OCH_3_), 56.2 (3″-OCH_3_/5″-OCH_3_), 32.0 (C-3), 31.4 (C-2), 28.9 (C-1).

##### 1,2,3-Trimethoxy-5-(3-(2,4,6-trimethoxyphenyl)propyl)benzene (**27**)

^1^H-NMR: (600 MHz, Me_2_CO-*d*_6_) δ: 6.42 (2H, s, H-2″/6″), 6.13 (2H, H-3′/5′), 3.84 (6H, s, 3‘’/5‘‘-OCH_3_), 3.82 (3H, s, 4′-OCH_3_), 3.81 (3H, s, 4′‘-OCH_3_), 3.79 (6H, s, 2‘/6‘-OCH_3_), 2.63 (2H, t, *J* = 7.4 Hz, H-1), 2.59 (2H, t, *J* = 7.4 Hz, H-3), 1.77 (2H, m, H-2); ^13^C-NMR: (150 MHz, Me_2_CO-*d*_6_) δ: 159.7 (C-2‘/4′/6‘), 153.5 (C-3″/5″), 138.5 (C-1′), 136.7 (C-4″), 111.8 (C-1‘), 105.6 (C-2‘/6‘), 90.8 (C-3‘/5‘),105.9 (C-2‘‘/6′‘), 60.8 (4‘/4″-OCH_3_), 56.0 (3″-OCH_3_/5″-OCH_3_), 55.7 (2′/6‘-OCH_3_), 36.4 (C-3), 30.7 (C-2), 22.4 (C-1).

##### 1,2,3-Trimethoxy-5-(3-(4-methoxyphenyl)propyl)benzene (**28**)

^1^H-NMR: (600 MHz, Me_2_CO-*d*_6_) δ: 7.11 (2H, d, *J* = 8.6 Hz, H-2′/6′), 6.84 (2H, d, *J* =8.6 Hz, H-3′/5′), 6.39 (2H, s, H-2″/6″), 6.68 (1H, brs, H-2″), 6.39 (2H, s, H-2′/6′), 3.90 (3H, s, 4‘’-OCH_3_), 3.85 (3H, s, 5‘’-OCH_3_), 3.83 (3H, s, 4′-OCH_3_), 3.79 (3H, s, 3‘‘-OCH_3_), 2.59 (2H, t, *J* = 7.4 Hz, H-3), 2.61 (2H, t, *J* = 7.4 Hz, H-1), 1.92 (2H, m, H-2); ^13^C-NMR: (150 MHz, Me_2_CO-*d*_6_) δ: 163.8 (C-4′), 153.5 (C-3″/5″), 140.1 (C-4″), 136.2 (C-1′), 131.0 (C-2‘/6‘), 115.0 (C-3‘/5‘), 112.0 (C-1‘), 105.9 (C-2″/6″), 61.0 (4″-OCH_3_), 56.4 (3″-OCH_3_/5″-OCH_3_), 55.6 (4′-OCH_3_), 36.4 (C-3), 30.7 (C-2), 22.4 (C-1).

##### 4-(3-(2,4,6-Trimethoxyphenyl)propyl)phenol (**29**)

^1^H-NMR: (600 MHz, CDCl_3_) δ: 7.06 (2H, d, *J* = 8.4 Hz, H-2″/6″), 6.73 (2H, d, *J* =8.4 Hz, H-3″/5″), 6.12 (2H, s, H-3′/5′), 3.80 (3H, s, 4′-OCH_3_), 3.78 (6H, s, 2′/6′-OCH_3_), 2.60 (2H, t, *J* = 7.5 Hz, H-1), 2.56 (2H, t, *J* = 7.9 Hz, H-3), 1.74 (2H, quin, H-2); ^13^C-NMR: (150 MHz, CDCl_3_) δ: 159.2 (C-4′), 159.0 (C-2‘/6‘), 153.4 (C-4″), 135.5 (C-1′), 129.8 (C-2‘‘/6′‘), 115.3 (C-3″/5″), 111.7 (C-1‘), 90.9 (C-3‘/5‘), 55.8 (2′-OCH_3_/6′-OCH_3_), 55.5 (4′-OCH_3_), 35.2 (C-3), 31.4 (C-2), 22.5 (C-1).

##### 3-(3-Hydroxy-3-(3,4,5-trimethoxyphenyl)propyl)phenol (**30**)

Compound **30** was obtained as a by-product during the synthesis of diarylpropane **26** in a yield of 25%. ^1^H-NMR: (600 MHz, CDCl_3_) δ: 7.15 (1H, t, *J* = 7.7 Hz, H-5″), 6.75 (1H, d, *J* = 7.6 Hz, H-6″), 6.67 (1H, d, *J* =7.7 Hz, H-4″), 6.68 (1H, brs, H-2″), 6.57 (2H, s, H-2′/6′), 4.61 (1H, m, H-1), 3.85 (6H, s, 3′-OCH_3_/5′-OCH_3_), 3.84 (3H, s, 4′-OCH_3_), 2.68 (2H, m, H-3), 2.09 (1H, m, H-2a), 1.98 (1H, m, H-2b); ^13^C-NMR: (150 MHz, CDCl_3_) δ: 155.8 (C-3‘’), 153.3 (C-3‘/5‘), 143.6 (C-1″), 140.3 (C-1′), 137.3 (C-4‘), 129.8 (C-5‘’), 120.9 (C-6‘’), 113.0 (C-4″), 103.1 (C-2‘/6′), 61.0 (4″-OCH_3_), 56.2 (3″-OCH_3_/5″-OCH_3_), 32.0 (C-3), 31.4 (C-2), 28.9 (C-1).

##### General Synthetic Procedure for Diarylpropenoic Acids (**31**–**42**)

Triethylamine (0.14 mL, 1.215 mmol) was added dropwise to a solution of the appropriate non-ortho-hydroxylated benzaldehyde (0.81 mmol) and phenyl acetic acid in acetic anhydrite (0.5 mL). The mixture was heated to 110 °C and stirred for 5–6 h. A solution of K_2_CO_3_ 10% (10 mL) was added and the reaction was left in reflux for 1–2 h. After cooling, the mixture was poured into an ice–water mixture and extracted with ethyl acetate. Hydrochloric acid was added to the aqueous phase until pH 3–4 and was extracted with ethyl acetate. The combined extracts were washed with water, brine, dried (Na_2_SO_4_) and evaporated in vacuo. The residue was purified by flash column chromatography on silica gel using c-Hexane/EtOAc as the eluents to afford the respective diarylpropenoic acids **31**–**42** (yield 78–89%) as solids. In the case of the ortho-hydroxylated benzaldehydes, the reaction afforded 3-arylcoumarins **48** and **49** as the main products (yields 58% and 60% respectively).

##### (*E*)-2-(4-Methoxyphenyl)-3-(4′-hydroxyphenyl)propenoic Acid (**31**)

White solid, yield: 83%; ^1^H-NMR: (600 MHz, Me_2_CO-*d*_6_) δ: 7.76 (s, 1 H, CH=), 7.16 (d, 2H, *J* = 8.6, 2/6-H), 7.01 (d, 2H, *J* = 8.6, 2′/6′-H), 6.95 (d, 2H, *J* = 8.6, 3/5-H), 6.67 (d, 2H, *J* = 8.6, 3′/5′-H), 3.82 (s, 3H, -OCH_3_-4); ^13^C-NMR: (150 MHz, Me_2_CO-*d*_6_) *δ*: 167.9 (COOH), 158.7 (4-C), 158.3 (4′-C), 139.8 (CH=), 132.8 (2′/6′-C), 131.7 (2/6-C), 130.8 (=CCOOH), 128.9 (1-C), 126 (1′-C), 115 (3′/5/-C), 114 (3/5-C), 55.5 (4-OCH_3_).

##### (*E*)-2-(Phenyl)-3-(4-hydroxyphenyl)propenoic Acid (**32**)

White solid, yield: 80%, ^1^H-NMR: (600 MHz, Me_2_CO-*d*_6_) δ: 7.79 (s, 1 H, CH=), 7.39–7.36 (m, 3H), 7.23 (dd, 2H, *J* = 8.6/2.1, 2/6-H), 6.97 (d, 2H, *J* = 8.6, 2′/6′-H), 6.65 (d, 2H, *J* = 8.6, 3′/5′-H); ^13^C-NMR: (150 MHz, Me_2_CO-*d*_6_) *δ*: 169.4 (COOH), 159.6 (4′-C), 140.1 (CH=); 158.7 (4-C), 132.8 (2′/6′-C), 131.7 (2/6-C), 130.8 (=CCOOH), 128.9 (1-C), 126 (1′-C), 115 (3′/5/-C), 114 (3/5-C), 55.5 (4-OCH_3_).

##### (*E*)-2- (4-Methoxyphenyl)-3-(3-hydroxyphenyl)propenoic Acid (**33**)

White solid, yield: 78%, ^1^H-NMR: (600 MHz, Me_2_CO-*d*_6_) δ: 7.90 (1H, t, *J* = 8.0 Hz, 5′-H), 7.74 (s, 1 H, CH=), 7.14 (2H, d, *J* = 8.6 Hz, 2/6-H), 6.92 (2H d, *J* = 8.6 Hz, 3/5-H), 6.73 (m, 2H, 2′/6′-H), 6.72 (1H dt, *J* = 8.0, 2.1 Hz, 4′-H), 3.82 (s, 3H, -OCH_3_-4); ^13^C-NMR: (150 MHz, Me_2_CO-*d*_6_) δ: 169.0 (COOH), 160.0 (4-C), 140.1 (CH=), 137.1 (1′-C), 133.2 (=CCOOH), 131.7 (2/6-C), 130.0 (5′-C), 128.6 (1-C), 122.5 (6′-C), 117.5 (2′-C), 116.7 (4′-C), 115.0 (3′-C), 114.0 (3/5-C), 55.5 (4-OCH_3_).

##### (*E*)-2-(4-Βromophenyl)-3-(4-hydroxyphenyl)propenoic Acid (**34**)

White solid, yield: 78%, ^1^H-NMR: (600 MHz, Me_2_CO-*d*_6_) δ: 7.83 (s, 1 H, CH=), 7.56 (2H, d, *J* = 8.3 Hz, 2/6-H), 7.14 (2H d, *J* = 8.3 Hz, 3/5-H), 6.95 (d, 2H, *J* = 8.6 Hz, 2′/6′-H), 6.62 (2H, d, *J* = 8.6 Hz, 3′/5′-H); ^13^C-NMR: (150 MHz, Me_2_CO-*d_6_*) δ: 167.9 (COOH), 158.3 (4′-C), 141.0 (CH=), 132.8 (2′/6′-C), 131.7 (2/6-C), 130.8 (=CCOOH), 128.9 (1-C), 127.6 (1′-C), 122.7 (4-C), 115.0 (3′/5′-C), 114.0 (3/5-C).

##### (*E*)-2-(4-Fluorophenyl)-3-(4-hydroxyphenyl)propenoic Acid (**35**)

White solid, yield: 83%, ^1^H-NMR (600 MHz, Me_2_CO-*d*_6_) δ: 7.81 (s, 1 H, CH=), 7.26 (2H, d, *J* = 8.6 Hz, 2/6-H), 7.17 (2H d, *J* = 8.6 Hz, 3/5-H), 6.99 (d, 2H, *J* = 8.6 Hz, 2′/6′-H), 6.69 (2H, d, *J* = 8.6 Hz, 3′/5′-H); ^13^C-NMR (150 MHz, Me_2_CO-*d*_6_) δ: 167.9 (COOH), 162.7 (4-C), 158.3 (4′-C), 141.0 (CH=), 132.8 (2′/6′-C), 131.7 (2/6-C), 130.8 (=CCOOH), 128.9 (1-C), 126.0 (1′-C), 115.0 (3′/5′-C), 114.0 (3/5-C).

##### (*E*)-2-(4-Chlorophenyl)-3-(4-hydroxyphenyl)propenoic Acid (**36**)

White solid, yield: 89%, ^1^H-NMR (600 MHz, Me_2_CO-*d*_6_) δ: 7.83 (s, 1 H, CH=), 7.45 (2H, d, *J* = 8.6 Hz, 2/6-H), 7.28 (2H d, *J* = 8.6 Hz, 3/5-H), 6.95 (d, 2H, *J* = 8.6 Hz, 2′/6′-H), 6.72 (2H, d, *J* = 8.6 Hz, 3′/5′-H); ^13^C-NMR (150 MHz, Me_2_CO-*d*_6_) δ: 167.9 (COOH), 158.3 (4′-C), 141.3 (CH=), 132.8 (2′/6′-C), 131.7 (2/6-C), 130.8 (=CCOOH), 128.9 (1-C), 126.0 (1′-C), 122.7 (4-C), 115.0 (3′/5′-C), 114.0 (3/5-C).

##### (*E*)-2-(3,4-Dihydroxyphenyl)-3-(4-hydroxyphenyl)propenoic Acid (**37**)

White solid, yield: 78%, ^1^H-NMR (600 MHz, Me_2_CO-*d*_6_) δ: 7.70 (s, 1 H, CH=), 7.05 (2H, d, *J* = 8.6 Hz, 2′/6′-H), 6.84 (1H, d, *J* = 8.2 Hz, 5-H), 6.71 (d, 1H, *J* = 2.1 Hz, 2-H), 6.68 (2H, d, *J* = 8.6 Hz, 3′/5′-H), 6.55 (dd, 1H, *J* = 8.2 /2.1 Hz, 6-H); ^13^C-NMR (150 MHz, Me_2_CO-*d*_6_) δ: 169.4 (COOH), 158.3 (4′-C), 156.1 (4-C), 154.0 (3-C), 139.1 (CH=), 132.8 (2′/6′-C), 131.7 (2/6-C), 131.1 (=CCOOH), 128.9 (1-C), 121.5 (5-C), 115.0 (3′/5′-C).

##### (*E*)-2-(Phenyl)-3-(3-hydroxyphenyl)propenoic Acid (**38**)

White solid, yield: 80%, ^1^H-NMR (600 MHz, Me_2_CO-*d*_6_) δ: 7.82 (s, 1 H, CH=), 7.25 (2H, dd, *J* = 8.6, 2.1 Hz, 2/6-H), 7.36 (1H, s, 4-H), 7.02 (1H, t, *J* = 8.0 Hz, 5′-H), 6.73 (1H, dt, *J* = 8.0, 2.1 Hz, 4′-H), 6.60 (2H, m, 2′/6′-H); ^13^C-NMR (150 MHz, Me_2_CO-*d*_6_) δ: 169.0 (COOH), 159.0 (3′-C), 141.5 (CH=), 136.8 (1′-C), 132.0 (=CCOOH), 130.7 (2/6-C), 130.6 (5′-C), 129.4 (4-C), 129.2 (3/5-C), 128.9 (1-C), 122.8 (6′-C), 118.0 (2′-C), 117.3 (4′-C). 

##### (*E*)-2-(3-Hydroxy-4-methoxyphenyl)-3-(3-hydroxyphenyl)propenoic Acid (**39**)

White solid, yield: 80%, ^1^H-NMR (600 MHz, Me_2_CO-*d*_6_) *δ*: 7.72 (s, 1 H, CH=), 7.05 (1H, t, *J* = 8.0 Hz, 5′-H), 6.91 (1H, d, *J* = 8.9 Hz, 5-H), 6.87 (1H, dd, *J* = 8.9, 2.0 Hz, 6-H), 6.82 (1H, d, *J* = 2.0 Hz, 2-H), 6.73 (1H, dt, *J* = 8.0, 2.1 Hz, 4′-H), 6.62 (2H, m, 2′/6′-H), 3.82 (s, 3H, -OCH_3_-4); ^13^C-NMR (150 MHz, Me_2_CO-*d*_6_) δ: 169.0 (COOH), 159.0 (3′-C), 154.0 (3-C), 141.5 (CH=), 136.8 (1′-C), 131.7 (2/6-C), 131.4 (=CCOOH), 131.0 (5′-C), 128.9 (1-C), 123.0 (6′-C), 121.5 (5-C), 118.3 (2′-C), 118.0 (4′-C), 55.5 (OCH_3_).

##### (*E*)-3-(4-Hydroxy-3,5-dimethoxyphenyl)-2-phenylpropenoic Acid (**40**)

White solid, yield: 80%, ^1^H-NMR (600 MHz, Me_2_CO-*d*_6_) δ: 7.75 (s, 1 H, CH=), 7.45 (2H, m, 2/6-H), 7.36 (1H, t, *J* = 7.6 Hz, 3/5-H), 7.28 (2H, m, 3/5-H), 6.44 (2H, s, 2′/6′-H), 3.54 (s, 6H, -3′/5′-OCH_3_); ^13^C-NMR (150 MHz, Me_2_CO-*d*_6_) δ: 169.0 (COOH), 158.0 (5′-C), 157.0 (3′-C), 154.0 (4′-C), 141.3 (CH=), 136.2 (1-C), 132.0 (=CCOOH), 131.7 (2/6-C), 131.0 (2′/6′-C), 129.3 (1′-C), 128.0 (3/5-C), 127.8 (4-C), 57.5 (3′/5′-OCH_3_).

##### (*E*)-3-(4-Hydroxy-3,5-dimethoxyphenyl)-2-(4-methoxyphenyl)propenoic Acid (**41**)

White solid, yield: 86%, ^1^H-NMR (600 MHz, Me_2_CO-*d*_6_) δ: 7.73 (s, 1 H, CH=), 7.18 (2H, d, *J* = 8.6 Hz, 2/6-H), 7.01 (2H, d, *J* = 8.6 Hz, 3/5-H), 6.46 (2H, s, 2′/6′-H), 3.82 (s, 3H, -OCH_3_-4), 3.56 (s, 6H, 3′/5′-OCH_3_); ^13^C-NMR (150 MHz, Me_2_CO-*d_6_*) δ: 169.0 (COOH), 160.0 (4-C), 158.0 (5′-C), 157.0 (3′-C), 154.0 (4′-C), 141.5 (CH=), 132.1 (=CCOOH), 131.7 (2/6-C), 131.0 (2′/6′-C), 129.3 (1′-C), 128.6 (1-C), 114.0 (3/5-C), 57.5 (2′/5′-OCH_3_), 56.0 (OCH_3_). 

##### (*E*)-3-(3-Hydroxyphenyl)-2-(4-nitrophenyl)propenoic Acid (**42**)

White solid, yield: 78%, ^1^H-NMR (600 MHz, Me_2_CO-*d*_6_) δ: 8.27 (2H, d, *J* = 8.7 Hz, 2/6-H), 7.90 (s, 1 H, CH=), 7.54 (2H d, *J* = 8.7 Hz, 3/5-H), 7.09 (1H, t, *J* = 7.0 Hz, 5′-H), 6.77 (d, 1H, *J* = 7.0 Hz, 4′-H), 6.65 (1H, d, *J* = 7.0 Hz, 6′-H), 6.56 (1H, brs, 2′-H); ^13^C-NMR (150 MHz, Me_2_CO-*d_6_*) δ: 167.9 (COOH), 148.0 (4-C), 141.5 (CH=), 132.0 (=CCOOH), 131.7 (2/6-C), 130.6 (5′-C), 129.6 (1-C), 122.8 (1′-C), 122.3 (4′-C), 118.1 (6′-C), 117.0 (2′-C), 115.0 (3′-C), 114.0 (3/5-C).

##### General Synthetic Procedure for Diarylpropanoic Acids (**43**–**47**)

A solution of the appropriate diarylpropenoic acid (1mmol) in EtOAc (10 mL) was hydrogenated (Pd/C 10%, 120 mg) for 4–5 h at room temperature and 54 psi pressure. The catalyst was filtered out through celite, washed with AcOEt, and the combined filtrates were evaporated in vacuo. Purification of the residue by column chromatography on silica gel using c-Hexane/EtOAc provided the pure corresponding diarylpropanoic acids **43**–**47** (yields 82–92%).

##### 2-(4-Methoxyphenyl)-3-(4-hydroxyphenyl)propanoic Acid (**43**)

White solid, yield: 92%, ^1^H-NMR (600 MHz, Me_2_CO-*d*_6_) δ: 7.24 (2H, d, *J* = 8.6 Hz, 2/6-H), 6.94 (2H, d, *J* = 8.6 Hz, 2′/6′-H), 6.86 (2H d, *J* = 8.6 Hz, 3/5-H), 6.61 (2H, d, *J* = 8.6 Hz, 3′/5′-H), 3.77 (1 H, t, *J* = 7.8 Hz, C*H*), 3.28 (1H, dd, *J* = 13.8, 8.5 Hz, 8a-H), 2.90 (1H, dd, *J* = 13.8, 7.0 Hz, 8b-H); ^13^C-NMR (150 MHz, Me_2_CO-*d_6_*) δ: 168.0 (COOH), 160.0 (4-C), 158.3 (4′-C), 137.1 (1′-C), 132.8 (2′/6′-C), 131.7 (2/6-C), 128.6 (1-C), 115.1 (3/5-C), 115.0 (3′/5′-C), 54.0 (CH-), 40.7 (CH_2_).

##### 2-Phenyl-3-(4-hydroxyphenyl)propanoic Acid (**44**)

White solid, yield: 82%, ^1^H-NMR (600 MHz, Me_2_CO-*d*_6_) δ: 7.33–7.29 (4H, m, 2/3/5/6-H), 7.24 (1H, m, 4-H), 6.97 (2H, d, *J* = 8.8 Hz, 2′/6′-H), 6.65 (2H, d, *J* = 8.8 Hz, 3′/5′-H), 3.79 (1 H, t, *J* = 7.8 Hz, C*H*), 3.28 (1H, dd, *J* = 13.9, 8.7 Hz, 8a-H), 2.91 (1H, dd, *J* = 13.9, 7.0 Hz, 8b-H); ^13^C-NMR (150 MHz, Me_2_CO-*d_6_*) δ: 168.0 (COOH), 156.4 (4′-C), 137.1 (1′-C), 133.5 (2′/6′-C), 131.0 (4-C), 128.6 (1-C), 126.4 (2/3/5/6-C), 116.1 (3′/5′-C), 54.1 (CH-), 41.0 (CH_2_).

##### 2-(4-Methoxyphenyl)-3-(3-hydroxyphenyl)propanoic Acid (**45**)

White solid, yield: 86%, ^1^H-NMR (600 MHz, Me_2_CO-*d*_6_) δ: 7.24 (2H, d, *J* = 8.6 Hz, 2/6-H), 7.11 (1H, t, *J* = 8.0 Hz, 5′-H), 6.86 (2H d, *J* = 8.6 Hz, 3/5-H), 6.63 (1H, dt, *J* = 8.0, 2.0 Hz, 4′-H), 6.61 (1H, t, *J* = 2.0 Hz, 2′-H), 6.60 (1H, dt, *J* = 8.0, 2.0 Hz, 6′-H), 3.77 (1 H, ~t, *J* = 8.0 Hz, C*H*), 3.28 (1H, dd, *J* = 13.8, 8.5 Hz, 8a-H), 2.90 (1H, dd, *J* = 13.8, 7.0 Hz, 8b-H); ^13^C-NMR (150 MHz, Me_2_CO-*d*_6_) δ: 168.0 (COOH), 160.0 (4-C), 158.0 (3′-C), 137.1 (1′-C), 130.3 (2/6/5′-C), 128.6 (1-C), 121.4 (4′-C), 117.0 (2′-C), 114.2 (6′-C), 114.0 (3/5-C), 54.0 (CH-), 40.7 (CH_2_).

##### 2-(4-Bromophenyl)-3-(4-hydroxyphenyl)propanoic Acid (**46**)

White solid, yield: 90%, ^1^H-NMR: (600 MHz, Me_2_CO-*d*_6_) δ: 7.60 (2H, d, *J* = 8.3 Hz, 2/6-H), 7.10 (2H d, *J* = 8.3 Hz, 3/5-H), 6.93 (d, 2H, *J* = 8.6 Hz, 2′/6′-H), 6.60 (2H, d, *J* = 8.6 Hz, 3′/5′-H); 3.78 (1 H, t, *J* = 7.8 Hz, C*H*), 3.28 (1H, dd, *J* = 13.8, 8.5 Hz, 8a-H), 2.89 (1H, dd, *J* = 13.8, 7.0 Hz, 8b-H); ^13^C-NMR: (150 MHz, Me_2_CO-*d*_6_) δ: 169.3 (COOH), 158.0 (4′-C), 132.8 (2′/6′-C), 131.0 (2/6-C), 129.0 (1-C), 127.6 (1′-C), 122.7 (4-C), 115.0 (3′/5′-C), 114.0 (3/5-C), 54.2 (CH-), 41.0 (CH_2_).

##### 2-(4-Fluorophenyl)-3-(4-hydroxyphenyl)propanoic Acid (**47**)

White solid, yield: 92%, ^1^H-NMR (600 MHz, Me_2_CO-*d*_6_) δ: 7.35 (2H, d, *J* = 8.3 Hz, 2/6-H), 7.01 (2H d, *J* = 8.3 Hz, 3/5-H), 6.96 (2H, d, *J* = 8.6 Hz, 2′/6′-H), 6.66 (2H, d, *J* = 8.6 Hz, 3′/5′-H), 3.80 (1 H, t, *J* = 7.8 Hz, C*H*), 3.27 (1H, dd, *J* = 13.8, 8.5 Hz, 8a-H), 2.83 (1H, dd, *J* = 13.8, 7.0 Hz, 8b-H); ^13^C-NMR (150 MHz, Me_2_CO-*d*_6_) δ: 169.1 (COOH), 162.7 (4-C), 158.3 (4′-C), 132.9 (2′/6′-C), 130.7 (2/6-C), 128.9 (1-C), 126.0 (1′-C), 115.4 (3′/5′-C), 114.0 (3/5-C), 54.1 (CH-), 40.7 (CH_2_).

##### 7-Hydroxy-3-(4-hydroxyphenyl)-2*H*-chromen-2-one (**48**)

White solid, yield: 58%, ^1^H-NMR (600 MHz, Me_2_CO-*d*_6_) δ: 7.98 (1H, s, 4-H), 7.56 (2H d, *J* = 8.5 Hz, 5-H), 7.56 (2H, d, *J* = 8.7 Hz, 2′/6′-H), 6.85 (2H, d, *J* = 8.7 Hz, 3′/5′-H), 6.82 (1H, dd, *J* = 8.5, 2.5 Hz, 6-H), 6.76 (1 H, d, *J* = 2.5 Hz, 8-H); ^13^C-NMR (150 MHz, Me_2_CO-*d_6_*) δ: 163.5 (2-C), 162.5 (7-C), 159.2 (4′-C), 156.3 (8a-C), 141.0 (4-C), 131.0 (2′/6′-C), 130.7 (5-C), 128.1 (3-C), 116.2 (3′/5′-C), 113.7 (1′-C), 103.2 (8-C), 102.8 (4a-C), 95.9 (6-C).

##### 7-Methoxy-3-phenyl-2*H*-chromen-2-one (**49**)

White solid, yield: 60%, ^1^H-NMR (600 MHz, Me_2_CO-*d*_6_) δ: 7.76 (1H, s, 4-H), 7.65–7.35 (5H, m, 2′/3′/4′/5′/6′-H), 7.01 (2H d, *J* = 8.7 Hz, 5-H), 6.81 (1H, dd, *J* = 8.7, 2.7 Hz, 6-H), 6.32 (1 H, d, *J* = 2.7 Hz, 8-H), 3.88 (s, 3H, -OCH_3_-7); ^13^C-NMR (150 MHz, Me_2_CO-*d_6_*) δ: 162.5 (7-C), 160.8 (2-C), 155.3 (8a-C), 141.0 (4-C), 135.0 (1′-C), 130.0–128.4 (2′/3′/4′/5′/6′-C), 129.0 (5-C), 124.8 (3-C), 113.4 (4a-C), 112.0 (6-C), 108.0 (8-C), 56.0 (OCH_3_-7).

#### 2.1.3. Biological Studies

##### Mushroom Tyrosinase Assay

The enzyme assay was performed by using mushroom tyrosinase and L-DOPA, as a substrate purchased from Sigma-Aldrich Chemical Co. (Burlington, MA, United States). The enzyme, the substrate and all of the samples were diluted in phosphate-buffered saline (pH 6, 7). The evaluation was performed in a 96-well plate and the wells were divided into four groups containing the following: (A) 120 μL phosphate buffer (1/15 M; pH 6, 7) and 40 μL mushroom tyrosinase (92 units/mL in phosphate buffer), in triplicate, as the control; (B) 160 μL phosphate buffer, in one well, as a blank of the control; (C) 80 μL phosphate buffer, 40 μL of sample dissolved in phosphate buffer (containing up to 3% DMSO) and 40 μL mushroom tyrosinase (92 units/mL in phosphate buffer), in triplicate; (D) 120 μL phosphate buffer and 40 μL of sample dissolved in phosphate buffer (containing up to 3% DMSO), in one well, as a blank of the sample. The contents of each well were incubated for 10 min at room temperature, before 40 μL of L-DOPA (2.5mM in phosphate buffer) were added. After incubating at room temperature for 5 min, the absorbance at 475 nm was measured. Kojic acid (KA2) and Glycyrrhiza glabra (Gly5) methanolic extract from the roots were used as the positive control. The percentage of inhibition of tyrosinase activity was calculated by the following equation:% inhibition of Tyrosinase = ((OD_A-OD_B) − (OD_C-OD_D))/(OD_A − OD_B) × 100

##### Cell Lines and Cell Culture Conditions

MTT Cytotoxicity Assay

Mouse skin melanoma B16F1 and B16F10 cells were obtained from the American Tissue Culture Collection (Manassas, VA, USA). Both skin melanoma cell lines were cultured at 5000 cells/well. After 24 h cells were treated with different concentrations of the compounds (in control cells, an appropriate amount of DMSO was added). The MTT dye (1 mg/mL in phenol-red-free DMEM w/o FBS) was added 48 h after the addition of the compounds. The reduction of the dye by the living cells was allowed to take place for 3–4 h. The MTT solution was discarded, and isopropanol was added to dissolve the formazan crystals. The absorbance of the solution was measured at 570 nm wavelength. The survival of the non-treated cells was set to 100%.

Melanin Content Assay

Melanoma cells were cultured at 5 × 10^5^ cells/plate. After 24 h, the cells were treated with the compounds (in control cells, an appropriate amount of DMSO was added); in all cases the compounds under study were applied at a concentration that was equal to their IC50. After 48 h, the cells were washed with PBS and were harvested by trypsinization. The cells were centrifuged, and photographs of the cell pellets were taken to reveal whitening of cells (this was mostly evident in the B16F10 cell line). Cell pellets were then solubilized in 200 μL of 1 M NaOH and were left at 95 °C for 1 h. The absorbance was measured at 405 nm using a spectrophotometer. Relative melanin content was calculated after setting the values obtained from non-treated cells to 100%.

Cellular Tyrosinase Activity Assay

The cells were plated at a density of 25 × 10^3^ cells/plate in 96-well plates. After 24 h, the cells were treated with the test substances (in control cells, an appropriate amount of DMSO was added); as for the melanin assay, the compounds were applied at concentrations equal to their IC50. After 48 h, the cells were washed with cold PBS and lysed with phosphate buffer (pH 6.8) containing 1% triton-X/PBS (90 μL/well). The cells were then left frozen at −80 °C for 30 min. After thawing and mixing, 100 μL of 0.1% L-DOPA was added to each well followed by sample incubation at 37 °C for 1 h. The absorbance was then measured at 492 nm using a spectrophotometer.

#### 2.1.4. In Silico Studies

##### Protein Preparation

The protein (PDB entry 2Y9X) was prepared for the docking calculations using the Protein Preparation Workflow (Schrödinger Suite 2021 Protein Preparation Wizard) implemented in Schrödinger suite and accessible from within the Maestro program (Maestro, version 12.8, Schrödinger, LLC, New York, NY, USA, 2021). Briefly, the hydrogen atoms were added and the orientation of hydroxyl groups, Asn, Gln, and the protonation state of His were optimized to maximize hydrogen bonding. Finally, the ligand−protein complex was refined with a restrained minimization performed by Impref utility, which is based on the Impact molecular mechanics engine (Impact version 9.1, Schrödinger, LLC, New York, NY, USA, 2021) and the OPLS_2005 force field, setting a max rmsd of 0.30. Ligand preparation for docking was performed with the LigPrep (Schrödinger, LLC, New York, NY, USA, 2021) application which consists of a series of steps that perform conversions, apply corrections to the structure, generate ionization states and tautomers, and optimize the geometries. 

##### Ligand Preparation

All ligands were designed using Maestro software (Maestro, version 12.8, Schrödinger, LLC, New York, NY, USA, 2021). Furthermore, LigPrep (Schrödinger, LLC, New York, NY, USA, 2021) was used to generate tautomeric, stereochemical, and ionization variations for all ligands. Finally, partial charges were predicted from the force field OPLS2005.

##### Docking Simulations

The induced-fit docking algorithm was utilized for molecular docking as implemented on Schrödinger Suite 2021. For calculating the grid box size, the center of the grid box was taken to be the center of the ligand in the crystal structure, and the length of the grid box for the receptor was twice the distance from the ligand center to its farthest ligand atom plus 10 Å in each dimension. The scoring calculations were performed using standard precision (SP).

#### 2.1.5. Free-Radical-Scavenging Assays

##### DPPH-Scavenging Assay

The DPPH antioxidant capacity of extracts was determined based on the Lee et al. protocol [16], scaled down for application in a 96-well plate reader (Table 2). A stock solution of 0.105 mM DPPH (Merck, Darmstadt, Germany) in absolute ethanol (Merck, Darmstadt, Germany) was prepared. This stock solution was prepared daily, used for the measurements, and kept in the dark at ambient temperature when not being used. Stock solutions of samples in DMSO (10 mg/mL) were prepared and dilutions were made in the testing concentration in the same solvent. A total of 10 μL of extract (200 µg/mL final concentration in the well) in DMSO and 190 μL of DPPH solution were mixed and incubated for 30 min in 96-well plates at ambient temperature in the dark. Absorbance was measured at 517 nm in a TECAN microplate reader. Those extracts that exhibited an inhibitory activity of more than 70% were measured again at lower concentrations. Blanks for every sample w/o DPPH were also measured. Ascorbic acid was used as the positive control. The percentage of DPPH scavenging was estimated by the following equation: ((A − B) − (C − D))/(A − B) × 100, where A: control (w/o sample), B: blank (w/o sample, w/o DPPH), C: sample, D: blank sample (w/o DPPH).

##### ABTS-Scavenging Assay

A stock solution of 2,2′-azino-bis (3-ethylbenzothiazoline-6-sulphonic acid) (ABTS) was prepared one day before the assay as follows: 10 mL of 7 mM ABTS solution were mixed with 164 µL of 140 mM of potassium persulfate and stored overnight (~16 h) at ambient temperature. The final solution after storage was diluted in H_2_O (1:20) until reaching an absorbance of 0.70 ± 0.02 (100 µL of ABTS solution plus 50 µL of DMSO). A total of 100 µL of the final ABTS solution and 50 µL of the sample (200 µg/mL final concentration in the well) were mixed in a 96-well plate. Those extracts that exhibited an inhibitory activity of more than 70% were measured again at lower concentrations. The plates were incubated at ambient temperature in the dark for 10 min and absorbance was measured at 734 nm in a TECAN microplate reader. Blanks for every sample w/o ABTS were also measured. Trolox was used as a positive control (8 µg/mL final concentration). The percentage of ABTS scavenging was estimated by the following equation: ((A − B) − (C − D))/(A − B) × 100, where A: control (w/o sample), B: blank (w/o sample, w/o ABTS), C: sample, D: blank sample (w/o ABTS) (Table 2).

## 3. Results and Discussions

### 3.1. Design and Synthesis of Compounds

Chalcones consist of two phenyl rings (A and B), one α, β-unsaturated double bond and a ketone (middle three-carbon chain), and show structural analogy to the previously described deoxybenzoins [17]. Most importantly, they are the precursors of diarylopropanes which, compared to the active dihydrostilbenes, are differentiated by an extra carbon in the middle chain. Based on the above, we proceeded to the design and synthesis of 30 chalconoid analogues with various substitution patterns. Chalcones (*E-*derivatives) (**1**–**15**) were obtained via a base-catalyzed condensation of a mixture of substituted acetophenones and benzaldehydes in alcohol (Claisen–Schmidt reaction) [18], which upon catalytic hydrogenation [19] of the double bond, provided the corresponding diarylopropanes (**18–29**). The hydrogenation of chalcone **13** also afforded the alcohol derivative (**30**) as a by-product. The dihydrochalcones (**16**–**17**) were synthesized via a Friedel–Crafts acylation [20] between properly substituted alcohol and phenyl acetic acid moieties. The synthetic route is illustrated in Figure 2.

As a next step, we decided to synthesize and investigate the possible activity of diarylpropenoic acids due to their structural similarities to stilbenes, which are well-known tyrosinase inhibitors [21]. Moreover, via a simple reduction, they can afford diarylpropanoic acids, which only differ from dihydrostilbenes in the existence of a carboxylic group in the middle chain. So, 12 diarylpropenoic acids (**31**–**42**) were prepared according to the Perkin condensation between substituted phenyl acetic acids and non-ortho-hydroxylated benzaldehydes, predominantly providing the (*E*) isomers with a cis relationship of the phenyl rings [22]. However, in the case of ortho-hydroxylated benzaldehydes, the reaction specifically afforded 3-arylcoumarins (**48**–**49**), and almost no *E*-diarylpropenoic acids were obtained. Probably, the appropriate distance between the ortho-hydroxyl and carboxyl groups led to an intramolecular cyclization of the *E*-diarylpropenoic acid intermediate, affording the 3-arylcoumarins, which were more energetically and geometrically favorable [23]. The catalytic hydrogenation of diarylpropenoic acids was applied in order to reduce the double bond and obtain the corresponding diarylpropanoic acids [24] (**43**–**47**) (Figure 3).

All of the synthesized compounds are depicted in Figure 1 and were fully characterized by spectroscopic methods. Previous studies from our group revealed some preliminary information on the structure–activity relationships regarding tyrosinase activity. The current evaluation granted the variety of scaffolds and substitution patterns of the synthesized compounds, contributing to the determination of the structural requirements for optimal activity.

### 3.2. Tyrosinase-Inhibition Properties

All of the synthetic compounds were screened for their tyrosinase-inhibition properties in the concentration of 100 μM. The tested compounds were characterized as strong (70–100%), moderate (40–70%) and weak tyrosinase inhibitors (20–40%) according to their inhibition percentage. The compounds that exhibited a negative inhibition percentage were considered as tyrosinase activators and were characterized/categorized accordingly.

The majority of chalcones did not inhibit tyrosinase activity except for derivatives **3**, **10** and **11,** which were characterized as weak inhibitors. Diarylpropanes **19** and **28** exhibited weak inhibition properties, whereas derivatives **24** and **25** were proved to be strong inhibitors, hampering tyrosinase activity up to 98%. It is worth noticing that both compounds have the resorcinol moiety in ring A, which was combined to increased inhibitory activity as previously described. Furthermore, compared to their chalcone precursors, the elimination of the carbonyl group seemed to significantly increase tyrosinase inhibition. The negative contribution of the carbonyl group to tyrosinase inhibition is further implied by the weak inhibitory properties of dihydrochalcone **16**. Compound **17** also proved to be a weak tyrosinase inhibitor, whereas its diarylpropane derivative **29** enhanced tyrosinase activity by about 50%. The alcohol derivative **30** was characterized as a potent tyrosinase inhibitor, hampering tyrosinase activity up to 76%. From the 2,3-diarylpropenoic acid group, only compound **37** exhibited weak inhibition while the rest of the derivatives exhibited either marginal enzyme inhibition or activation, with derivatives **38** and **39** enhancing tyrosinase activity up to 50%. Their reduced derivatives, 2,3-diarylpropanoic acids, exhibited exclusively activating properties. More specifically, compounds **43** and **45** were characterized as weak activators and compounds **46** and **47** as moderate activators, whereas derivative **44** proved to be a strong tyrosinase activator (−73% inhibition) with significantly enhanced activity compared to its precursor (compound **32**). The tyrosinase activity of all the synthetic compounds and the known inhibitor kojic acid is depicted in Figure 2.

### 3.3. Molecular Simulations Studies

#### Molecular Docking on Mushroom Tyrosinase

In order to obtain better insight into the mechanism by which the synthetic analogues interact with tyrosinase and modulate its activity, in silico studies of selected compounds were performed. Diarylpropane **24** was chosen as a representative strong inhibitor, and compounds **38** and **44** were selected as representative activators in order to investigate the structure–activity relationships.

Docking calculations using the crystal structure of tyrosinase predicted two binding modes of diarylpropane **24** in the active site of the enzyme (Figure 3) and provided an explanation of the in vitro results. In the first binding mode, OH-2 forms HB with N260 and E256, OH-4 interacts with catalytic copper (electrostatic interaction), and OH-4′ forms HB with N260 as well (Figure 3A,C). In the second binding mode, two hydrogen bonds (HB) were formed between OH-2 of the phenyl ring and the side chains of N260, H244, OH-4 with E256 and OH-4′ of the second phenyl ring with E322 (Figure 3B,D). Probably, the internal HB that formed between OH-2 and the carbonyl group in the cases of its precursors, chalcone **11** and dihydrochalcone **17**, reduced their binding affinity and thus their biological potency.

Furthermore, the binding modes of compounds **38** and **44** were examined using flexible docking simulations (Figure 4). In the absence of a ligand inside the binding cavity, all three compounds were bound in the active site (Figure 4A) with the carboxylic group forming a salt bridge with the copper ions of the protein. This could lead to hypothesizing that the aforementioned compounds could be potent tyrosinase inhibitors. However, in the presence of the crystallographic ligand (Tropolone) (Figure 4B), diarylpropenoic acids **38** (blue) and **44** (orange) bound in a different position, which could enhance the binding of tyrosine, the crystallographic ligand of the protein. Presumably, the formation of hydrogen bonds between the compounds and amino acids outside the active site of the enzyme, as well as the π–π stacking interactions between the protein and the analogues, stabilized the receptor’s structure, resulting in either an accelerated reaction or a better binding of the substrate, thereby enhancing enzyme activity. This is in accordance with the results from the in vitro evaluation that also indicated tyrosinase activation, since the calculated percentage of tyrosinase inhibition was negative.

### 3.4. Cytotoxicity and Determination of Melanin Content and Cellular Tyrosinase Activity

Four representative compounds were further investigated in murine melanoma cell lines, B16F1 and B16F10, which intracellularly synthesize melanin and then secrete it into the extracellular culture fluid. The B16F10 cell line, compared to B16F1, produces higher amounts of melanin and is characterized by high metastatic potential and resistance to chemotherapy [25]. Due to the different activities of the tested compounds, both cell lines were used for the evaluation. More specifically, 2,3-diarylpropanoic acid **44** was chosen as a potent activator along with its precursor 2,3-diarylpropenoic acid **32**, while the previously described 2,4,4′-trihydroxy-dihydrostilbene **I** along with the 2,4-dihydroxy-dihydrostilbene **II** were selected as potent tyrosinase inhibitors. The cytotoxicity of the synthetic compounds as well as their ability to alter tyrosinase activity and extracellular melanin content in B16F10 and B16F1 melanoma cells were measured. The compounds showed marginal toxicity to both of the tested cell lines, and their IC_50_ values are presented in Table 1. All compounds were tested in their IC_50_ concentrations.

Regarding the effect of the synthetic analogues on tyrosinase activity (Figure 5), 4-hydroxy-2,3-diphenylpropenoic acid (**32**) did not affect tyrosinase activity in the B16F1 cells; however, in B16F10 it reduced the enzymatic activity by 18%. Its reduced analogue, 4-hydroxy-2,3-diphenylpropanoic acid (**44**), enhanced tyrosinase activity in both of the cell lines and by up to 15% in B16F1 cells. On the other hand, 2,4,4′-trihydroxy-dihydrostilbene **I** (Figure 1) was characterized as the most potent inhibitor, hampering tyrosinase activity by about 23% in the B16F1 cell line and 63% in the B16F10 cells (which was statistically significant), compared to the control (CTRL). Finally, 2,4-dihydroxy-dihydrostilbene **II** (Figure 1) inhibited tyrosinase activity by 17% in the B16F1 cells and 50% in B16F10, proving to be a more potent inhibitor than kojic acid. All of the obtained results are in accordance with the in vitro cell-free evaluation.

Regarding melanin production, 4-hydroxy-2,3-diphenylpropenoic acid (**32**), even though it did not suppress tyrosinase activity in the B16F1 cells, suppressed melanin production by up to 25%. While statistically significant, it proved less active in the B16F10 cells. The 4-hydroxy-2,3-diphenylpropanoic acid (**44**) significantly enhanced melanin production by 33% in the B16F1cell line following the tyrosinase activation, but surprisingly, in the B16F10 cells it suppressed melanin production by 20% in relation to the control and it was statistically significant. 2,4,4′-trihydroxy-dihydrostilbene **I** (Figure 1) revealed the greatest melanin reduction of 30% in the B16F1 and 72% in the B16F10 cell lines compared to the control (statistically significant). Finally, 2,4-dihydroxy-dihydrostilbene **II** (Figure 1) reduced melanin production by up to 56% in the B16F10 cell line (statistically significant) and up to 26% in the B16F1 cells, showing comparable activity to kojic acid. Photographs of the cell pellets (centrifuged cells) of the B16F1 and B16F10 cell lines after treatment with the synthetic analogues are presented in Figure 6, revealing the whitening or blackening effect of the tested compounds.

### 3.5. Free-Radical-Scavenging Properties

Five chalcones (**2**, **11**, **13**–**15**), four diarylpropanes (**18**, **22**, **26**, **29**), seven diarylpropenoic acids (**31**–**33**, **35**, **39**–**41**), two diarylpropanoic acids (**45**, **47**) and the dihydrochalcone **17** were selected based on their structure and evaluated for their free-radical-scavenging properties against DPPH and ABTS (Table 2). The results showed that twelve of the abovementioned compounds (**11**, **17**, **26**, **29**, **31**, **33**, **35**, **39**–**41**, **45**, **47**) possessed significant ABTS-scavenging activity as their IC50 values (12.4–68.5 μΜ) were lower than that of ascorbic acid (122.1 μΜ). Furthermore, two of the evaluated diarylpropenoic acids (40 and 41) showed scavenging activity (IC50 values 16.1 and 14.1 μΜ, respectively) similar to that of ascorbic acid (IC50 value 24.1 μΜ). As was expected, the compounds without a phenolic hydroxyl group (**2**, **14**, **15**, **22**) lacked the free-radical-scavenging ability. 

## 4. Conclusions

In summary, 49 compounds were synthesized, providing a variety of scaffolds and substituents, and were evaluated in vitro for their mushroom-tyrosinase-inhibitory activity and their free-radical-scavenging properties. The obtained results showed weak tyrosinase-inhibitory properties only for the hydroxy chalcones **10** and **11**, due to their resorcinol moiety, whereas the intermediate carbonyl group seemed to hamper the inhibitory activity. The reduction of the α, β-unsaturated double bond in the case of dihydrochalcones did not affect the activity; instead, the elimination of the ketone group enhanced the inhibitory properties in the case of diarylpropanes **24** and **25**, proving them to be more potent inhibitors than kojic acid. The diarylpropanol derivative **30** also emerged as a potent inhibitor, probably due to the existence of the free OH group of the middle chain. The majority of diarylpropenoic acids enhanced tyrosinase activity, whereas their reduced analogues—the diarylpropanoic acids—revealed improved activating properties, with compound **44** emerging as the most potent tyrosinase activator. Docking simulation studies of diarylpropane **24** indicated the necessity of free hydroxyl groups that are able to interact with the amino acids in the active site of the enzyme in order to exhibit strong tyrosinase inhibition. On the other hand, conformational analyses of compound **44** revealed that the stabilizing interactions outside the active site of the enzyme could enhance tyrosinase activity. Moreover, the cell-based evaluation using B16F1 and B16F10 melanoma cells demonstrated that dihydrostilbene analogues **I** and **II** exhibited a stronger anti-melanogenic effect than kojic acid through the inhibition of cellular tyrosinase activity and melanin formation. On the contrary, diarylpropanoic acid **44** proved to be a potent melanogenic factor, inducing cellular tyrosinase activity and melanin formation. Finally, the antioxidant activity assays revealed compounds **29** and **11** as significant free-radical-scavenging agents. Specifically, they exhibited 10- and 6-fold more potent scavenging activity than ascorbic acid, respectively. Overall, the above compounds could be considered as safe and promising candidates for the development of novel therapeutic agents or cosmeceuticals for dermatological or neurological disorders that are associated with melanin pigments and free radicals.

## Data Availability

Data is contained within the article or Appendix A.

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
