# Peer review of "Development of Stilbenoid and Chalconoid Analogues as Potent Tyrosinase Modulators and Antioxidant Agents"

_antioxidants, 2022, doi:10.3390/antiox11081593_

Round 1

Reviewer 1 Report

lines 688-690: Docking calculations using the crystal structure of tyrosinase predicted two binding modes of diarylpropane 24 in the active site of the enzyme (Figure 3) and provided an explanation of the in vitro results. 

3.3.2. Ligand Based – Machine Learning Classification and Regression

Did the authors use all 277 molecular descriptors for generating the SVM classification model? What about the risk of overfitting? Did the authors cross-validate the SVM model? What was the out-of-sample misclassification error? Is the SVM model reliable to "be further utilized as filter in future tyrosinase inhibitors"?

In my opinion, subsection 3.3.2. could be omitted, or more data should be presented.

Author Response

Comments and Suggestions for Authors

  • lines 688-690: Docking calculations using the crystal structure of tyrosinase predicted two binding modes of diarylpropane 24 in the active site of the enzyme (Figure 3) and provided an explanation of the in vitro results.

The above sentence has been revised according to the reviewer's comment 

  • 3.2. Ligand Based – Machine Learning Classification and Regression

Did the authors use all 277 molecular descriptors for generating the SVM classification model? What about the risk of overfitting? Did the authors cross-validate the SVM model? What was the out-of-sample misclassification error? Is the SVM model reliable to "be further utilized as filter in future tyrosinase inhibitors"?

In my opinion, subsection 3.3.2. could be omitted, or more data should be presented.

The subsection 3.3.2. has been omitted

Reviewer 2 Report

Authors reported several chemical analogues on tyrosinase activity and melanin synthesis. This manuscript is very interesting.

However, some data are not clear, and author should perform experiment repetitively, especially melanin synthesis. 

Please check the following comment. 

Line 511 5x10”5” cells. Need to change to superscript.

               Also line 522.

Line 517 I recommend to use molar “M”. 

Table 1 is poor. Please reconstruct the table. 

Figure 5 Top left figure: the number of compounds are absent.

               Lower left figure: error bars are absent. 

I understand authors did only one time for melanin content assay using B16F1 cells. 

I recommend to do at least 3 experiments. 

Figure 6 is very unclear. For example, please take photo on the white paper. 

Author Response

Comments and Suggestions for Authors

Authors reported several chemical analogues on tyrosinase activity and melanin synthesis. This manuscript is very interesting.

However, some data are not clear, and author should perform experiment repetitively, especially melanin synthesis. 

Please check the following comment.

  • Line 511 5x10”5” cells. Need to change to superscript. Also line 522.

The lines 511 and 522 have been changed

  • Line 517 I recommend to use molar “M”.

The line 517 has been changed

  • Table 1 is poor. Please reconstruct the table.

The Table 1 has been reconstructed

  • Figure 5 Top left figure: the number of compounds is absent. Lower left figure: error bars are absent. I understand authors did only one time for melanin content assay using B16F1 cells. I recommend to do at least 3 experiments.

The experiments for melanin content assay using B16F1 cells were repeated three times in total and the figure 5 has been revised according to the reviewer’ comments

  • Figure 6 is very unclear. For example, please take photo on the white paper.

The figure 6 has been revised according to the reviewer’ comment

Reviewer 3 Report

The paper entitled "Development Of Stilbenoid And Chalconoid Analogues As Potent Tyrosinase Modulators and Antioxidant Agents" is well conceived and the methodology is well supported by the interesting results. It is my opinion that the work can be published as is.

Author Response

Comments and Suggestions for Authors

The paper entitled "Development Of Stilbenoid And Chalconoid Analogues As Potent Tyrosinase Modulators and Antioxidant Agents" is well conceived and the methodology is well supported by the interesting results. It is my opinion that the work can be published as is.

We would like to thank reviewers for their comments

Round 2

Reviewer 2 Report

I feel this manuscript can accept for "antioxidant".